# Surface texture limits transfer of *S. aureus*, T4 bacteriophage, influenza B virus and human coronavirus

**Qi Liu, Lindsey Brookbank, Angela Ho, Jenna Coffey, Anthony B. Brennan, Christopher J. Jones***

Sharklet Technologies, Inc. Aurora, CO, United States of America

\* cjones@sharklet.com

## Abstract

Spread of pathogens on contaminated surfaces plays a key role in disease transmission. Surface technologies that control pathogen transfer can help control fomite transmission and are of great interest to public health. Here, we report a novel bead transfer method for evaluating fomite transmission in common laboratory settings. We show that this method meets several important criteria for quantitative test methods, including reasonableness, relevancy, resemblance, responsiveness, and repeatability, and therefore may be adaptable for standardization. In addition, this method can be applied to a wide variety of pathogens including bacteria, phage, and human viruses. Using the bead transfer method, we demonstrate that an engineered micropattern limits transfer of *Staphylococcus aureus* by 97.8% and T4 bacteriophage by 93.0% on silicone surfaces. Furthermore, the micropattern significantly reduces transfer of influenza B virus and human coronavirus on silicone and polypropylene surfaces. Our results highlight the potential of using surface texture as a valuable new strategy in combating infectious diseases.

## Introduction

The spread of diseases requires the transmittance of pathogens from an infected host or carrier to a naive individual. There are many mechanisms for this to occur, including direct transfer of bodily fluids, airborne droplets, or indirect contact through a contaminated intermediate object, known as fomite transmission. Fomite transmission involves transfer of liquid between human subjects and environmental surfaces. In this mechanism, the pathogen is deposited through body secretion or aerosolization onto an object, such as a touch screen, hand rail, or desk, where it persists until it is acquired by a susceptible host through direct contact, usually by touching the object with their hands. The new host then transports the pathogen to a port of entry, such as a break in the skin, or the mucous membranes found in the eyes, nose, or mouth. Fomite transmission has been shown to occur with bacteria, fungi, and viruses [1–4], and has been associated with community outbreaks [5–8]. For example, one publication estimates that fomite transmission accounts for up to 85% of indoor transmission [9]. Therefore, technologies and sanitation practices that limit fomite transmission can be a useful tool in combating many disease outbreaks [10, 11].

**Data Availability Statement:** All relevant data are within the manuscript.

**Funding:** This work was funded by internal Sharklet Technologies, Inc. funding. All authors were employees of Sharklet Technologies, Inc.

while contributing to this study. The funder provided support in the form of salaries for authors [QL, LB, AH, JC, AB, and CJ] and test materials, but did not have any additional role in the study design, data collection and analysis, decision to publish, or preparation of the manuscript. The specific roles of these authors are articulated in the 'author contributions' section.

**Competing interests:** I have read the journal's policy and the authors of this manuscript have the following competing interest: all authors were employed by Sharklet Technologies while contributing to this manuscript. Additionally, Anthony B. Brennan is a member of the Sharklet Technologies, Inc. Board of Directors. This does not alter our adherence to PLOS ONE policies on sharing data and materials.

Strict sanitation and disinfection regimens have been implemented in a wide variety of public spaces, including schools, transportation, food production, and healthcare settings in an attempt to thwart fomite transmission of pathogens. These cleaning and disinfection procedures are often the first line of defense against the spread of pathogens. Many standards exist regarding the cleaning, disinfection, and sampling of surfaces, along with acceptable limits for contamination in these settings [12]. For example, Salgado et al. found that the there is a significantly increased risk for acquiring a Hospital Acquired Infection (HAI) in hospital rooms where surface bacterial burden exceeds 500 CFU/ 100cm$^2$ [13]. These cleaning and disinfection protocols are generally effective, especially when consistent surveillance programs are in place to identify problem areas. In spite of such efforts, occasionally outbreaks of disease highlight the gaps in the current cleaning and disinfection practice.

Another approach to limiting fomite transmission is to utilize single-use or disposable items, such as gloves and gowns. This is an effective approach in high-risk areas. However, there are concerns about waste, sustainability, cost, and resource demands. As the recent coronavirus pandemic has highlighted, reliance on disposable items like personal protective equipment can be shaken by disruptions to supply chains.

A third, and relatively new approach, is to modify surfaces to combat fomite transmission. The most common technique for surface modification has been the addition of a biocide to the material. Another method is to place a coating on the surface that either kills pathogens upon contact or prevents their adhesion. Metals (such as copper and silver), antibiotics, and other chemical disinfectants (e.g. hypochlorite) are commonly used. These additives have been shown to be highly successful, but often raise safety concerns around their toxicity and are susceptible to reduced efficacy over time due to biocide stability and leaching from the surface.

Rather than instill an additive or coating, a newer strategy to surface modification is to alter the structure of the surface. Often, these modifications are small in stature and inspired by textures found in nature. Examples of this type of biomimicry are gecko feet [14], lotus petals [15, 16], and insect wings [17–19]. These include microstructures to increase friction, alter hydrophobicity, or reduce bacterial adhesion, respectively.

One example of biomimicry is a series of engineered micropatterns based on the shark skin, called Sharklet® (SK) micropattern. Shark skin has been known for its excellent self-cleaning and anti-fouling properties, and microscopic analysis revealed that it consists of diamond-shaped scales called denticles. Each shark denticle has a series of protruding ridges, and SK micropattern was designed to mimic the denticles. The micropattern consists of 7 small bars in a diamond shape, with each bar having a width of 2–5 μm and a depth of 2–5 μm (Fig 1A). This texture can be applied to nearly any polymeric article and does not modify the composition of the base material. Previous research has demonstrated that SK micropattern is effective at reducing the transfer, migration, and contamination of a variety of surfaces by microbes including algae, bacteria, and fungi [20–24].

It has been proposed that SK micropattern reduces pathogen transmission as a three-part mechanism. First, the micropattern increases the hydrophobicity of the surface, and therefore reduces the amount of fluid transferred onto the surface during contact inoculation. Second, the micropattern presents a reduced surface area for contact by pathogens, reducing the strength of attachment and persistence on surfaces. Third, after initial contact, the action of drying pulls pathogens into the base of the pattern via capillary action, limiting the transfer of these pathogens during a subsequent touch. Collectively, these mechanisms reduce transfer of pathogens, such as *Staphylococcus aureus* by up to 2 logs [23].

Many viral pathogens are of great concern for public health, including but not limited to: adenovirus (respiratory infections), rhinovirus (common cold), respiratory syncytial virus (acute pulmonary infections in children), influenza (respiratory infection), norovirus

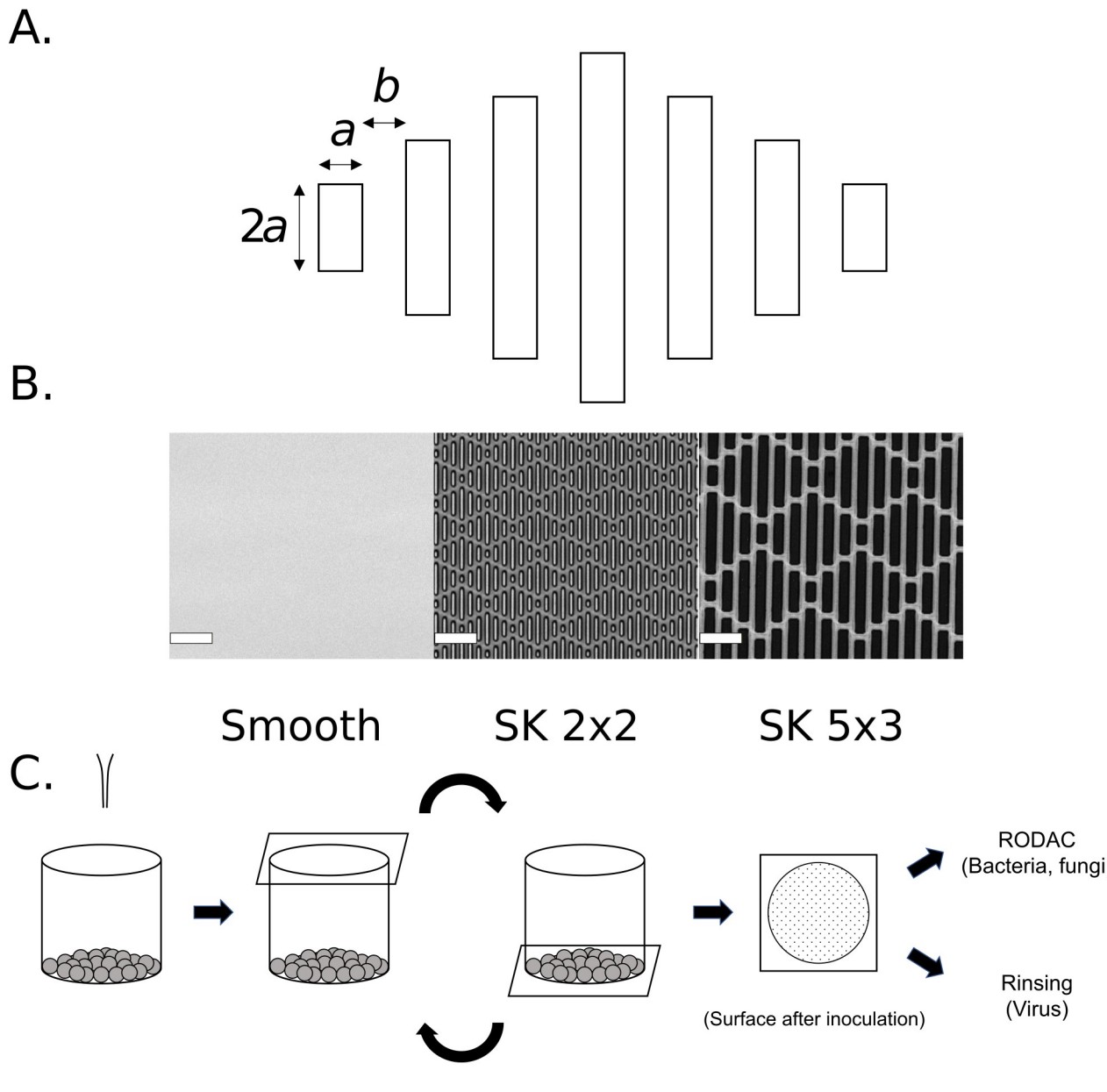

**Fig 1. Sharklet® micropattern and the bead transfer method.** (A) Schematic design of micropattern SK *a* x *b* and their fabrication on silicone surface. The repeating diamond-shaped micropattern consists of seven rectangular bars. The bars are *a* μm wide, 2*a*, 4*a*, 6*a*, or 8*a* μm long, and spaced b μm apart. For example, SK2x2 consists of rectangular bars that are 2 μm x 4 μm, 2 μm x 8 μm, 2 μm x 12 μm, and 2 μm x 16 μm, which are spaced 2 μm apart. (B) Representative confocal images are shown for smooth control, SK2x2, and SK5x3 silicone surfaces; scale bar, 20μm. (C) Schematic workflow for the bead transfer method. 1) For each sample, 15 grams of glass beads are dispensed into a sample cup and coated with inoculum immediately before testing. 2) Sterilized testing sample is placed over the sample cup with the testing surface facing down, and the entire apparatus is inverted three times to allow inoculation of the surface with glass beads. 3) After inoculation, the surface is either left air dry before sampling with RODAC plates (bacteria or fungi) or sampled immediately by rinsing (viruses).

(gastrointestinal infections), and the re-emergence of the measles virus (systemic infection). The impact of viral pathogens has been highlighted by several recent epidemics, such as SARS-CoV-1, MERS, Ebola, and the most recent pandemic strain, SARS-CoV-2. Viruses can survive on surfaces for days to weeks, which enhances fomite transmission, especially in areas that encourage close contact such as airplanes or schools [1]. Viral infections often exacerbate other health problems by forming secondary infections in conjunction with existing bacterial or fungal pathogens [25–27]. From a health-care perspective, limiting viral infections may have a greater impact on human health than limiting bacterial infections. One report found that more cases of community acquired pneumonia are attributed to viral agents (84/156, 54%) than bacterial agents (72/156, 46%) [28]. Despite the needs to control viral fomite transmission, few surface technologies have been shown for such efficacy.

Here, we describe the development of a novel technique to contaminate surfaces by utilizing glass beads coated in pathogens. This mimics fomite transmission of these organisms and can be applied to a wide variety of two-dimensional or three-dimensional objects. This technique was used to demonstrate the efficacy of SK micropattern in reducing the transmission of pathogens. The SK micropattern significantly reduces the transfer of model bacteria, phage, and human viruses. The techniques presented here can be used to quantify the reduction of fomite transmission with nearly any surface.

## Materials and methods

### Silicone sample generation

Silicone samples used for testing were casted inside a clean room using Elastosil M 4641 (Wacker). Elastosil part A was mixed with Elastosil part B at a 10:1 ratio by weight in a Speed-Mixer DAC600.2 VAC-LR (FlackTek, Inc.) for one minute under pressure following the manufacturer's recommendation. Mixed silicone was poured onto a nickel master with desired topography (smooth or SK patterned) and covered with a glass plate to allow even spreading across the master. The thickness of silicone samples was controlled by 1mm-thick spacers taped around the nickel master. The casting apparatus was incubated at 65˚C with 20lb weights on top for 2 hours for silicone to cure. Silicone samples were removed from the master and cut into 50mm x 50mm squares using a razor blade. The fidelity of surface topography on silicone samples was verified by confocal microscopy (LEXT OLS4000, Olympus, Fig 1B). A summary of pattern dimensions is included in Table 1. The silicone samples were submerged in 70% ethanol for 10 seconds and placed pattern side up onto a 100mm x 15mm petri dish. Samples were UV sterilized for 15 minutes before used in experiments.

### Contact angle measurement

Three-phase water contact angle for each silicone surface was measured using a Ramé-hart contact angle goniometer (Model 250). Surfaces were rinsed with 95% ethanol and air dried

**Table 1. Measurement of micropattern dimensions.**

| Material | Designed Micropattern | Measurement[a] | | |
|---|---|---|---|---|
| | | Height (μm) | Width (μm) | Space (μm) |
| Silicone | SK2x2 | 3.05 ± 0.02 | 2.00 ± 0.01 | 1.95 ± 0.04 |
| | SK5x3 | 4.11 ± 0.04 | 4.83 ± 0.06 | 3.04 ± 0.03 |
| Polypropylene | SK2x2 | 2.55 ± 0.05 | 2.53 ± 0.03 | 1.50 ± 0.01 |

[a]Measurement of micropattern dimension is reported by Mean ± SEM (n = 9).

completely before measurement. A 10uL water droplet was deposited onto the surface using a micropipette, and the image of the drop was recorded and analyzed using the built-in DROP-image Advanced software. For micropatterned surfaces, the length of SK features was placed along the light path. Measurement was repeated 4–11 times with each type of surface, and each measurement gives two readings for contact angle, from the left and right sides of the water drop. A total of 8–22 contact angle readings were collected for each type of surface and used for data analysis.

## Strains and media

*Staphylococcus aureus* (ATCC 6538) was grown in tryptic soy broth (TSB, Criterion C7141) at 37˚C and 280 rpm for 16 hours. Before testing, bacteria were pelleted by centrifugation at 250 x g for 5 minutes. Supernatant was discarded and *S. aureus* was resuspended in phosphate buffer saline (PBS, HyClone SH30258.02). OD600 of bacterial suspension was measured using a BioWave CO8000 (BioChrom), and theoretical CFU per mL was calculated using empirically determined OD600-CFU relationship under identical growth condition (1 OD600 = 5.065 x $10^8$ CFU/mL). *S. aureus* was diluted to desired concentration (5 x $10^3$ CFU/mL) using PBS and was used within 30 minutes of preparation. For each independent assay, the inoculum was serially diluted and plated on tryptic soy agar to confirm its concentration falls within acceptable range (2.5 x $10^3$–1 x$10^4$ CFU/mL).

Escherichia coli B strain (ATCC 11303) was grown in Luria-Bertani (LB) broth (Lennox, Sigma Aldrich L3022) at 37˚C and 280 rpm for 16 hours before phage propagation and plaque assays. MDCK cells (ATCC CCL-34) and MRC-5 cells (ATCC CCL-171) were maintained in cell culture media [Dulbecco's Modified Eagle Medium (DMEM, Gibco 11995) supplemented with 100 U/mL penicillin, 100 μg/mL streptomycin (Gibco 15140–122), and 10% fetal bovine serum (FBS, Gibco 10082–147)] at 37˚C under 5% $CO_2$.

## Phage and virus propagation

Bacteriophage T4 (ATCC 11303-B4) was prepared as described before [29]. *E. coli* B strain was subcultured in LB broth supplemented with 1mM $CaCl_2$ and $MgCl_2$ at 37˚C and 280 rpm until mid-log phase (OD600 = 0.4–0.8). Bacteriophage T4 was added to the *E. coli* culture at an MOI of 0.003, and incubated with agitation for about five hours until the lysate is visibly clear. Phage lysate was clarified by centrifugation at 250 x g for 30 minutes followed by passing through a 0.22μm syringe filter (Millipore SLGV033RS). The resulting phage stock was quantified by plaque assay (see below) and stored at 4˚C for up to two months before use. Phage was diluted to desired concentration (2.0 x$10^6$ PFU/mL) using SM buffer (50mM Tris-HCl, pH 7.5, 100mM NaCl, 8mM $MgSO_4$) before testing and was used within 30 minutes of preparation. For each independent assay, the inoculum was serially diluted and quantified by plaque assay to confirm its concentration falls within acceptable range (1.0–5.0 x $10^6$ PFU/mL).

Influenza B virus (IBV, ATCC VR-295) was propagated essentially as described [30]. MDCK cells were grown in a T175 until 80–90% confluence. Growth media was aspirated off and cells were washed three times with PBS before infected with 2mL IBV in infectious PBS [iPBS, PBS supplemented with 0.3% bovine serum albumin (BSA, Fisher BP1600), 100 U/mL penicillin, 100 μg/mL streptomycin, 2 mM $MgCl_2$, and 1mM $CaCl_2$] at an MOI of 0.01. Virus was absorbed for 60 minutes at 37˚C, shaking every 15 minutes. At the end of adsorption, viruses were aspirated off, and cells were washed three times with influenza infection media [IM-flu, DMEM supplemented with 100 U/mL penicillin, 100 μg/mL streptomycin, 0.3% BSA, 0.1% FBS, 20mM N-2-hydroxyethylpiperazine-N-2-ethane sulfonic acid (HEPES, Gibco 15630–080), and 1 μg/mL TPCK-treated trypsin (Sigma Aldrich T1426)]. IBV infected cells

were incubated at 37°C under 5% $CO_2$ for 48–72 hours. When 90% of cells were detached from the flask, the supernatant containing IBV was harvested and clarified at 500 x g for 5 minutes. The supernatant was aliquoted and stored at -80°C, and the titer of IBV was determined by TCID50 assay in MDCK cells (see below). Virus stocks were diluted to approximately 1.0 x $10^6$ PFU/mL. IBV was thawed on ice and used undiluted for testing.

Human coronavirus 229E (ATCC VR-740) was propagated in MRC-5 cells similarly as for IBV. Cells were infected with 2mL coronavirus 229E in infectious DMEM (iDMEM, DMEM supplemented with 100 U/mL penicillin, 100 μg/mL streptomycin, 0.2% BSA) at an MOI of 0.01. After an hour of adsorption, hCoV was removed and CoV infection media (IM-CoV, DMEM supplemented with 100 U/mL penicillin, 100 μg/mL streptomycin, and 2% FBS) was added. Once cytopathic effects (CPE) reach 50% or more, cell culture flask was frozen at -80°C for an hour and partially thawed in a 37°C water bath. Freeze thaw was repeated two more times to facilitate cell lysis and virus release. After the last thawing step, cell lysate was clarified at 3,000 x g for 10 minutes at 4°C. The supernatant was aliquoted and stored at -80°C, and the titer of coronavirus 229E was determined by TCID50 assay in MRC-5 cells (see below). Virus stocks were diluted to approximately 1.0 x $10^5$ PFU/mL. Coronavirus was thawed on ice and used undiluted for testing.

**Bead transfer method and microbial recovery from testing surfaces.** For each testing sample, 15g of 3mm soda-lime glass beads (Walter Stern 100C) were dispensed into a sterile sample cup (Thermo Scientific 021038). Immediately before inoculation, an appropriate amount of inoculum (1mL for bacteria or phage, 500μL for human viruses) was added to the beads (Fig 1A). The sample cup was capped with a lid and shaken by hand vigorously in a circular motion for 10 seconds for even dispersal of inoculum on glass beads. The lid was removed, and a testing sample was placed face down over the sample cup opening. The testing sample was held firmly against the sample cup by hand while inverted three times to allow inoculation of the surface by glass beads.

For bacteria recovery, the inoculated surface was air dried completely at room temperature for 5 minutes. RODAC plate was stamped onto the surface for 5 seconds, applying about 400g of pressure ensuring there are no bubbles trapped between the two surfaces. RODAC plates were incubated at 37°C overnight. Plates that contain between 30–300 individual colonies were counted to determine CFU per sample.

For virus recovery, 5mL SM buffer (phage) or 2mL Infection Media (IM, human viruses) were immediately pipetted onto the inoculated sample and over the surface three times bottom to top in each of the four directions. Viruses recovered from each testing surface were quantified by plaque assay (phage) or TCID50 assay (human viruses).

## Phage plaque assay

Phage recovered from testing samples were diluted in LB broth for countable plaques. 1mL recovered phage at proper dilution (neat, 10x, or 100x), 1mL *E.coli* B strain overnight culture, and 3mL molten 0.7% LB Agar were combined in a 15 mL sterile conical tube, and the tube was inverted once slowly to mix without generating any bubbles. The mixture was carefully poured onto an 1% LB agar plate and evenly distributed by tilting. The plate was allowed to sit undisturbed at room temperature until the agar has solidified (about five minutes) and incubated overnight at 37°C to allow plaque formation. Plates that contain 30–300 individual plaques were counted to determine PFU/mL for each sample:

$$PFU/mL = plaques\ per\ plate \times volume\ plated\ in\ mL \times dilution\ factor$$

## TCID50 assay

Cells compatible with viruses to be quantified were grown to ~90% confluency in a T175. On the day of the experiment, cells were washed once with PBS and detached from the flask by incubating with 2mL Trypsin-EDTA (HyClone, SH3023602) at 37˚C under 5% $CO_2$. Cells were diluted to 50,000 cells/mL in cell culture media and aliquoted into 96-well plates at 100μL per well. Cells were incubated for 1–2 hours at 37˚C or until cells were visually adhered to the bottom before use. Virus samples were serially diluted in cold IM in 24-well plates. IBV was diluted 10, 30, 90, 270, 810, 2430, 7290, and 21870-fold; CoV 229E was diluted 3, 9, 27, 81, 243, 729, and 2187-fold. Cell culture media was removed from the 96-well plates and cells were washed once with 100μL IM before 100μL diluted viruses were added. 8–12 wells were used for each virus dilution. For each 96-well plate, at least one well was used as uninfected control, where 100μL IM was added instead of viruses. Cells were incubated at 37˚C under 5% $CO_2$ for 4–5 days or until CPE became stagnant. At the end of incubation, cells were fixed with 4% paraformaldehyde in PBS (Thermo Scientific J19943K2) and stained with 0.2% (w/v) crystal violet (Sigma-Aldrich V5265) in 20% methanol. CPE was recorded for each well, and TCID50 was calculated using Reed-Muench method [31].

## Statistical analysis

A linear mixed-effects model was used to fit the log transformed counts ($\log_{10}$CFUs, $\log_{10}$PFUs, or $\log_{10}$TCID50) or log reductions (LRs) from all independent assays performed on each species. Random effects included in the model were date of experiment and operator performing the assay. Therefore, the mixed-effects model estimated the between-date variance ($V_D$), between-operator variance ($V_O$), and within-date and operator variance (residual error, $V_R$). The model can be expressed as,

$$y_{ijk} = \mu + D_i + O_j + \varepsilon_{ijk}$$

where $y_{ijk}$ represents the experimental readouts ($\log_{10}$CFUs, $\log_{10}$PFUs, $\log_{10}$TCID50, or LRs) from the $i^{th}$ date, the $j^{th}$ operator, and the $k^{th}$ assay, μ represents the mean readout over all experiments and operators, $D_i$ represents the between-date variance, $O_j$ represents the between-operator variance, and $\varepsilon_{ijk}$ represents the within-experiment and operator variance (residual error). All linear mixed models were fit in RStudio v.1.2.5042 using R v.3.6.3 [32] and package lme4 [33]. Residual, scale-location, and quantile-quantile plots were used to check linearity, homoscedasticity, and normality assumptions of the linear mixed-effects model. The repeatability standard deviation (RSD) for each species tested was calculated by:

$$RSD = V_D + V_O + V_R$$

Comparison between different surfaces (Figs 2–6) were performed in Prism 8 (Graph-Pad) using the following workflow: 1) outliers were identified using ROUT method (Q = 1%) and excluded from data analysis; 2) normality of the sample distribution was tested at significance level of 0.05 using Anderson-Darling test, D'Agostino-Pearson omnibus normality test, Shapiro-Wilk normality test, and Kolmogorov-Smirnov normality test with Dallal-Wikinson-Lillie for P value; 3) since all sample distributions reported in this study passed all normality tests, t test (two samples) or ordinary one-way ANOVA (three or more samples) and multiple comparison test recommended by the software was used for comparison.

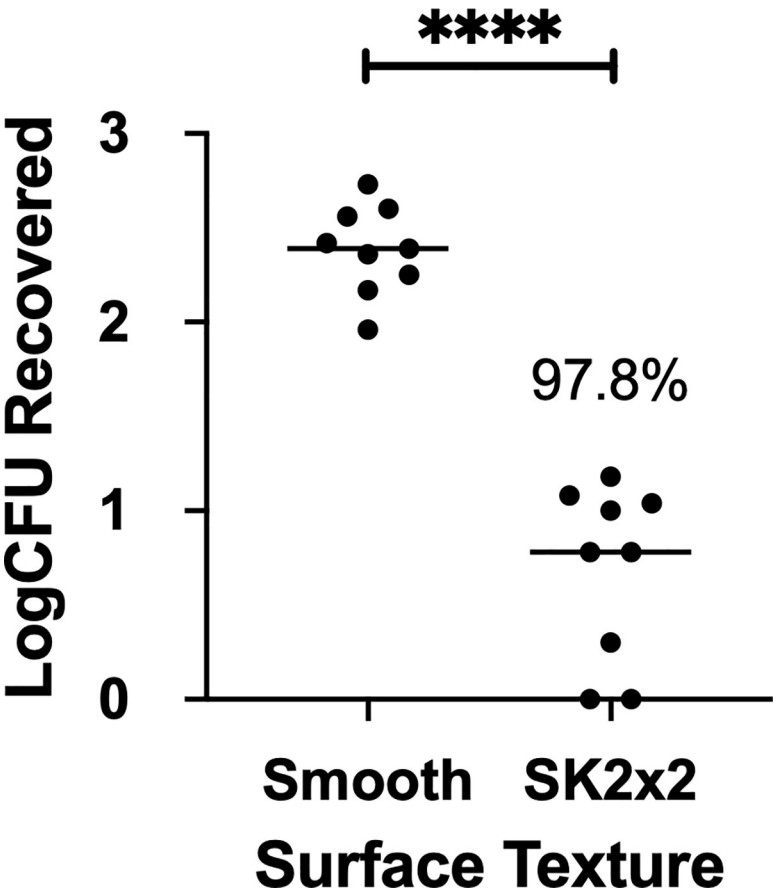

**Fig 2. SK2x2 micropattern limits transfer of *Staphylococcus aureus* on silicone surfaces.** Smooth or SK2x2 micropatterned silicone samples were tested against *Staphylococcus aureus* using bead transfer method. The log transformed colony forming units ($\log_{10}$CFUs) recovered per sample is indicated for each sample. Horizontal lines indicate sample mean. Percent reduction compared to smooth is indicated above the data set. Statistical significance was determined by unpaired two-tailed t test (****, $p < 0.0001$).

## Results

### The bead transfer method and *Staphylococcus aureus* testing

To develop a method for evaluating transfer of microorganism onto surfaces, we applied the following design criteria for the method: 1) is applicable for different types of microorganisms including bacteria, fungi, and viruses; 2) is applicable for a wide variety of surfaces, including three-dimensional surfaces that are commonly found in daily uses; 3) shows acceptable levels of attributes that are critical for standardized disinfectant tests, including reasonableness, relevancy, validity, ruggedness, resemblance, responsiveness, repeatability, and reproducibility [34]. It was reasoned that the bead transfer method (Fig 1C) could meet all these requirements and therefore was selected for further development.

For initial testing, the bead transfer method was used to evaluate attachment of *Staphylococcus aureus* on control (smooth) silicone. Briefly, glass beads were coated with *S. aureus* inoculum then applied to sterilized silicone surfaces (Fig 1C). After inoculation, tiny droplets are evenly distributed on the surface. The surface was sampled by a Replicate Organism Detection and Counting (RODAC) plate after it is visibly dry, and the amount of bacteria transferred onto the surface was determined by colony forming units (CFUs). To assess resemblance of

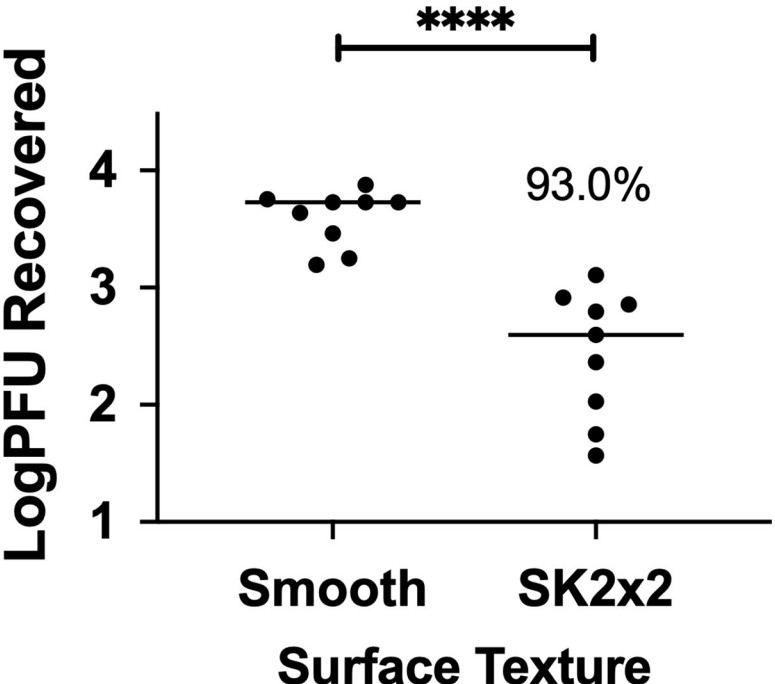

**Fig 3. SK2x2 micropattern limits transfer of Bacteriophage T4 on silicone surfaces.** Smooth or SK2x2 micropatterned silicone samples were tested against *Bacteriophage T4* using beads transfer method. The log transformed plaque forming units ($Log_{10}$PFUs) recovered per sample is indicated for each sample. Horizontal lines indicate sample mean. Percent reduction compared to smooth is indicated above the data set. Statistical significance was determined by unpaired two-tailed t test (****, $p < 0.0001$).

the method, i.e. the repeatability in control samples, this assay was repeated on two different days by three different operators (S1 Table). A linear mixed-effects model was used to fit the $\log_{10}$ transformed CFUs ($\log_{10}$CFU) from all independent assays, and the resemblance repeatability standard deviation (or control repeatability standard deviation, denoted by $CS_r$) was estimated to be 0.283 (Table 2). This value meets the historical acceptance criteria of $CS_r \leq 0.5$ [35], suggesting that the bead transfer method has reasonable resemblance for testing *S. aureus* attachment. Among the factors contributing to $CS_r$, it was estimated that 46.1% of the variance stems from between-date variation, and 32.1% results from between-operator variation.

Previously, we have demonstrated that an engineered Sharklet® micropattern, SK2x2, effectively reduces bacterial attachment to surfaces [20, 23, 24]. To test whether this micropattern is responsive to the bead transfer method, attachment of *S. aureus* on SK2x2 patterned silicone surface (Fig 1A, Table 1) was measured as described above (S1 Table). Compared to the smooth control, SK2x2 micropattern significantly reduces the amount of *S. aureus* transferred by beads, as shown by $\log_{10}$CFU (Fig 2, unpaired two-tailed t-test, $p < 0.0001$). For each independent assay, log reduction (LR) was calculated by subtracting $\log_{10}$CFU for patterned surface from $\log_{10}$CFU for control surface. Similarly, to account for between-date and between-operator variations, a linear mixed-effects model was used to fit LRs from all assays, in which LR was estimated to be 1.66 ± 0.24 (Table 3). In other words, SK2x2 micropattern results in an average reduction of *S. aureus* attachment on silicone surfaces by 97.8%. From the model, the repeatability standard deviation ($S_r$) was estimated to be 0.438 and meets the historical acceptance criteria of $S_r \leq 1.0$ [35], suggesting that the bead transfer method has reasonable repeatability for measuring the efficacy of antimicrobial surfaces against *S. aureus* attachment.

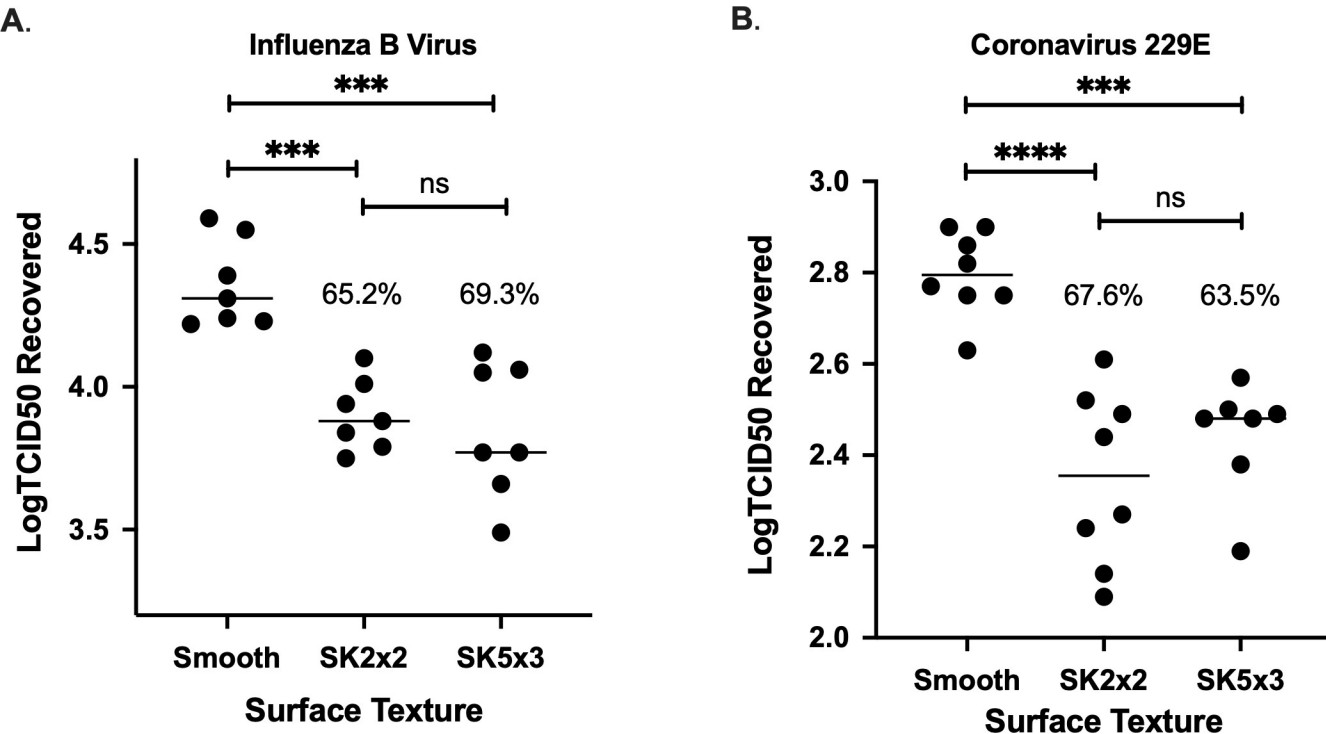

**Fig 4. SK micropattern limits transfer of human viruses on silicone surfaces.** Smooth, SK2x2, or SK5x3 micropatterned silicone samples were tested against *Influenzavirus B/Taiwan/2/62* (A) or *Coronavirus 229E* (B) using beads transfer method. The log transformed TCID50 (Log$_{10}$TCID50) recovered per sample is indicated for each sample. Horizontal lines indicate sample mean. Percent reduction compared to smooth is indicated above the data set. Statistical significance was determined by ordinary one-way ANOVA followed by Tukey's multiple comparison test (****, $p < 0.0001$; ***, $p < 0.001$; ns, not significant, $p > 0.05$).

### Transfer of bacteriophage T4

To investigate whether the bead transfer method can be applied for other types of microorganisms, we used this method to measure transfer of bacteriophage T4 on silicone surfaces. Inoculation of the surface was essentially the same as described above. To prevent desiccation which can be detrimental to phage survival, phage was recovered from the surface immediately after inoculation through extensive rinsing, and the amount of phage transferred on to the surface was quantified by plaque assay. Similarly, resemblance of the method for bacteriophage T4 was evaluated by performing the assay on three different days by three different operators (S2 Table). By fitting log transformed plaque forming units (log$_{10}$PFU) to a linear mixed-effects model, $CS_r$ was estimated to be 0.239, suggesting that the bead transfer method has reasonable resemblance for testing bacteriophage T4 transfer. Interestingly, the fitted model was (near) singular, where both between-date and between-operator variance was (nearly) zero (Table 2). This could be due to less biological variation between experiments, since phages are obligate parasites and do not metabolize by themselves.

To determine whether SK2x2 micropattern is effective against bacteriophage T4, the bead transfer method was repeated on patterned silicone surfaces (S2 Table). Compared to the smooth control, log$_{10}$PFU were significantly reduced by the SK2x2 micropattern (Fig 3, unpaired two-tailed t-test, $p < 0.0001$), indicating an inhibitory effect on bacteriophage T4 transfer. Account for between-date and between-operator variations, SK2x2 micropatterns resulted in an LR of 1.15 ± 0.25, or an average reduction of 93.0% in bacteriophage T4 transfer

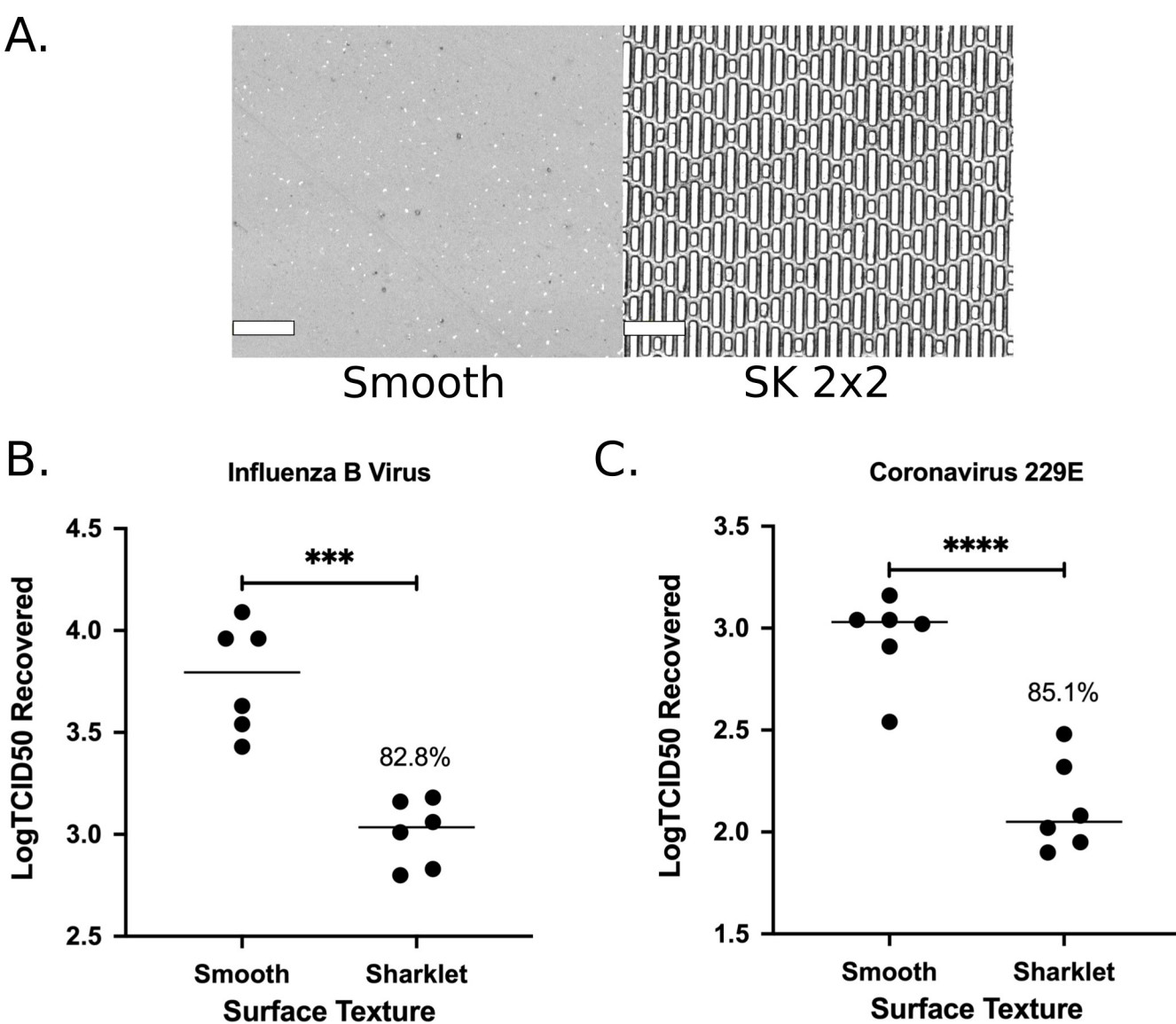

**Fig 5. SK micropattern reduces touch transfer of human viruses on Sharklet Shieldpolypropylene film.** (A) Representative confocal images of smooth control and SK2x2 polypropylene surfaces. Scale bar, 20μm. (B-C) Smooth or SK2x2 micropatterned polypropylene film was tested against *Influenzavirus B/ Taiwan/2/62* (B) or *Coronavirus 229E* (C) using beads transfer method. The log transformed TCID50 (Log$_{10}$TCID50) recovered per sample is indicated for each sample. Horizontal lines indicate sample mean. Percent reduction compared to smooth is indicated above the data set. Statistical significance was determined by unpaired two-tailed t test (****, $p < 0.0001$; ***, $p < 0.001$).

compared to the smooth control. $S_r$ for LRs was estimated to be 0.477 (Table 3), suggesting a reasonable level of repeatability for the assays.

### Transfer of influenza B virus on silicone surface

Virus transfer has been shown to vary among viral species [36–38], largely influenced by surface properties of viral particles [39]. In contrast to bacteriophage T4, where capsid proteins directly interact with inanimate surfaces, enveloped viruses have a membrane structure that shields viral particles from the outer environment. To determine whether the bead transfer method can reliably measure transfer of enveloped viruses to silicone surfaces, we applied this

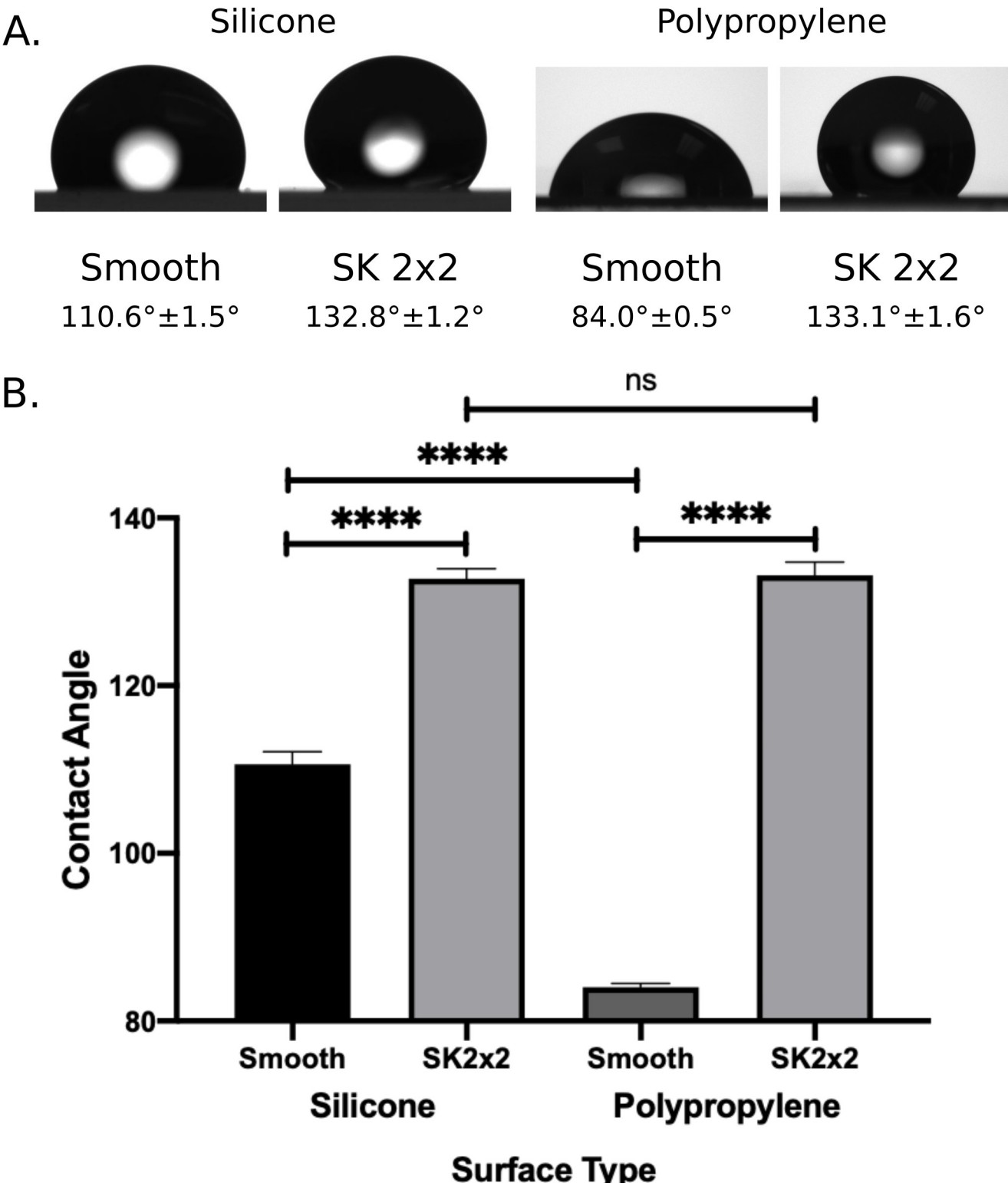

**Fig 6. Effects of SK micropatterns on surface wettability.** (A) Representative images of water drops captured by a goniometer on smooth and SK2x2 silicone or polypropylene surfaces. Water contact angles are reported belowimagesby mean ± SEM (silicone, n = 8; smooth polypropylene n = 16; SK2x2 polypropylene, n = 22). (B) Comparison of water contact angle on different surfaces, error bars represent SEM. Statistical significance was determined by ordinary one-way ANOVA followed by Holm-Sidak's multiple comparison test (****, $p < 0.0001$; ns, not significant, $p > 0.05$, $p > 0.05$).

**Table 2. Repeatability and variance components of the controls.**

| Species | Log$_{10}$ CFU/PFU/TCID50[a] | Repeatability SD ($CS_r$) | % Var: Date | % Var: Operator | % Var: Residuals |
|---|---|---|---|---|---|
| *Staphylococcus aureus* | 2.34 ± 0.17 | 0.283 | 46.1% | 32.1% | 21.8% |
| *Bacteriophage T4* | 3.60 ± 0.06 | 0.239 | 0% | 0% | 100% |
| *Influenza B* | 4.40 ± 0.09 | 0.169 | 0% | 49.9% | 50.1% |
| *Coronavirus 229E* | 2.80 ± 0.06 | 0.102 | 45.7% | 0% | 54.3% |

[a] Log$_{10}$CFU/PFU/TCID50 is reported by Mean ± SEM (*Staphylococcus aureus*, *Bacteriophage T4*, n = 9; *Influenza B*, n = 7; *Coronavirus 229E*, n = 8).

method to *Influenzavirus B/Taiwan/2/62*, an enveloped RNA virus. Independent assays were performed on two different days by three different operators using smooth silicone, and the amount of virus transferred was quantified by 50% tissue culture infective dose (TCID50) method in MDCK cells (S3 Table). Fitting log transformed TCID50 (log$_{10}$TCID50) to a linear mixed-effects model estimated $CS_r$ to be 0.169, suggesting that the bead transfer method has reasonable resemblance for influenza B virus (IBV). Between-operator variance contributes to 49.9% of $CS_r$, whereas between-date variance was (nearly) zero (Table 2).

Next, we examined the efficacy of SK micropatterns against transfer of IBV. In addition to SK2x2, we included a new generation SK topography, SK5x3, in our assays (S3 Table). In the SK5x3 micropattern, features are designed to be 4μm high, 5μm wide, spaced 3μm apart, and with length ranging from 10μm to 40μm (Fig 1A, Table 1). Previously, this new micropattern has been shown to limit bacterial touch transfer while exhibiting improved mechanical properties (internal proprietary data). When patterned silicone surfaces were subjected to the bead transfer method, both types of surfaces showed significant decrease in the amount of IBV transfer, as quantified by log$_{10}$TCID50s (Fig 4A, ordinary one-way ANOVA followed by Tukey's multiple comparison test, $p < 0.001$). SK2x2 and SK5x3 micropatterns resulted in LRs of 0.46 ± 0.06 and 0.51 ± 0.11, or average reduction of 65.2% and 69.3% in IBV transfer compared to the smooth surface, respectively (Table 3). There was no significant difference between LRs from SK2x2 and SK5x3 (unpaired two-tailed t-test, $p = 0.159$). $S_r$ for SK2x2 and SK5x3 was estimated to be 0.161 and 0.300, respectively, suggesting reasonable repeatability for both micropatterns.

## Transfer of coronavirus 229E on silicone surface

In light of the emerging pandemic of coronavirus disease 2019 (COVID-19), we were interested in whether SK micropatterns can be used for preventing the spread of coronaviruses. We

**Table 3. Repeatability and variance components of the LRs.**

| Species | Surface Type | LR[a] | Percentage Reduction (PR) | Repeatability SD ($S_r$) | % Var: Date | % Var: Operator | % Var: Residuals | PR Upper 95% CI[b] |
|---|---|---|---|---|---|---|---|---|
| *Staphylococcus aureus* | SK2x2 | 1.66 ± 0.24 | 97.8% | 0.438 | 20.1% | 51.9% | 28.0% | 94.2% |
| *Bacteriophage T4* | SK2x2 | 1.15 ± 0.25 | 93.0% | 0.477 | 28.7% | 49.0% | 22.3% | 81.1% |
| *Influenza B* | SK2x2 | 0.46 ± 0.06 | 65.2% | 0.161 | 0% | 0% | 100% | 55.8% |
| | SK5x3 | 0.51 ± 0.11 | 69.3% | 0.300 | 0% | 0% | 100% | 52.1% |
| *Coronavirus 229E* | SK2x2 | 0.49 ± 0.11 | 67.6% | 0.195 | 0% | 51.4% | 48.6% | 50.7% |
| | SK5x3 | 0.44 ± 0.11 | 63.5% | 0.092 | 0% | 0% | 100% | 42.1% |

[a] LR is reported by Mean ± SEM (*Staphylococcus aureus*, *Bacteriophage T4*, n = 9; *Influenza B*, n = 7; *Coronavirus 229E*, n = 8).
[b] PR upper 95% CI was calculated from LR upper 95% CI, which was computed in R using the default method in the stats package.

reason that the bead transfer method can be readily applied for evaluating transfer of coronaviruses, which are enveloped RNA viruses like influenza, on surfaces. As a proof-of-concept study, we used human coronavirus (hCoV) 229E, a surrogate strain for SARS-COV-2 as suggested by ASTM E35.15 [40]. Similarly, control (smooth) silicone was independently tested on two different days by two different operators, and surface contamination was quantified by the TCID50 method in MRC-5 cells (S3 Table). The resulting $\log_{10}$TCID50 was fitted into a linear mixed-effects model, which estimates $CS_r$ to be 0.102, confirming that the bead transfer method has reasonable resemblance for hCoV 229E. Between-date variance contributes to 45.7% of $CS_r$, whereas between-operator variance was (nearly) zero (Table 2).

To determine the efficacy of SK micropatterns against hCoV 229E, SK2x2 and SK5x3 patterned silicone was subject to the bead transfer method alongside with smooth control (S3 Table). As seen with influenza, both micropatterns significantly reduce hCoV transfer to silicone surfaces (Fig 4B ordinary one-way ANOVA followed by Tukey's multiple comparison test, SK2x2, $p < 0.0001$; SK5x3, $p < 0.001$). Accounting for between-date and between-operator variation, SK2x2 micropattern led to an LR of 0.49 ± 0.11, or an average reduction of 67.6% in hCoV transfer; SK5x3 micropattern led to an LR of 0.44 ± 0.11, or an average reduction of 63.5% in hCoV transfer (Table 3). There was no significant difference between LRs from SK2x2 and SK5x3 (unpaired two-tailed t-test, $p = 0.960$). From the linear mixed models used to fit the LRs, $S_r$ for SK2x2 and SK5x3 was estimated to be 0.195 and 0.092, respectively, indicating reasonable repeatability for both micropatterns. These results were highly reminiscent of those seen with IBV, suggesting that SK micropatterns have similar inhibitory effects towards the transfer of enveloped RNA viruses on surfaces.

## Sharklet® polypropylene film

Having established that the bead transfer method is suitable for evaluating surface contamination by human viruses, we applied this method for a commercially available product, Sharklet® polypropylene (PP) film, to assess its efficacy in limiting viral transfer. The PP film was produced by a partner manufacturer with SK2x2 micropattern on the front side, and smooth on the back side (Table 1, Fig 5A). Therefore, the back side of the film was used as the control surface. Six independent assays were performed by two different operators on two different days for both *Influenza virus B/Taiwan/2/62* and hCoV 229E (S4 Table). When PP film was challenged with IBV, SK2x2 micropattern resulted in a significant reduction in viral transfer (Fig 5B, unpaired two-tailed t-test, $p < 0.001$), with an LR of 0.76 ± 0.11, or an average reduction of 82.8%. Similarly, when tested with hCoV, SK2x2 micropatterned PP film showed a significant reduction in viral transfer (Fig 5C, unpaired two-tailed t-test, $p < 0.0001$) compared to the smooth control, with an LR of 0.83 ± 0.08, or an average reduction of 85.1%.

## SK micropattern alters the interaction of surfaces with water

To our surprise, for both IBV and hCoV, SK2x2 micropattern led to a significantly greater reduction in viral transfer on PP film compared to silicone (unpaired two-tailed t-test, IBV, $p = 0.0271$; hCoV, $p = 0.0018$). One mechanism of action for SK micropatterns is through lowering surface energy and therefore wettability of the surface. To investigate the difference in SK micropattern efficacy between different material, we measured water contact angle for smooth and SK2x2 micropatterned silicone and PP film (Fig 6). Interestingly, smooth PP film is significantly more wettable than smooth silicone (ordinary one-way ANOVA followed by Holm-Sidak's multiple comparison test, $p < 0.0001$), whereas SK2x2 micropatterned PP film showed similar wettability as SK2x2 micropatterned silicone ($p = 0.8751$). We reasoned that

the greater difference in wettability between smooth and micropatterned PP film may explain the elevated effects of SK2x2 micropattern on PP film compared to silicone.

## Discussion

Quantitative standardized methods are critical for evaluating the efficacy of disinfectants and antimicrobial surfaces. Despite their importance, there are very few standardized methods that can be readily adaptable for different types of microorganisms (bacteria, fungi, virus) and three-dimensional surfaces. Here we report for the first time an in-house developed bead transfer method and show that this method has adequate levels of resemblance, responsiveness, repeatability for bacteria, phage, and human viruses. In addition, we argue that the bead transfer method is both reasonable and relevant: the protocol is easy and inexpensive to implement, and liquid droplets transferred by glass beads can mimic a number of real-life scenarios such as touching common surfaces with dirty hands or contaminated equipment. Glass beads can be used to inoculate a variety of non-porous surfaces, including three-dimensional surfaces and objects, and inoculation efficiency may be optimized by adjusting bead size according to the surface geometry of interest. Future development of this method will involve testing of other microorganisms such as fungi, assessment of method ruggedness, as well as interlaboratory studies to determine reproductivity.

SK micropatterns have been shown to limit touch transfer and bioadhesion for a number of microorganisms including bacteria, fungi, and algae zoospores [20–24]. Here we show that these micropatterns are effective against transfer of viral particles as well. To the best of our knowledge, this is the first example of using surface topographies to control viral interactions with inanimate surfaces, which has become increasingly important in epidemiology due to the current pandemic of coronavirus disease. There are numerous advantages in using topographies to limit microbial transfer: 1) physical modification offers continuous protection of the surface; 2) there are no added chemicals during manufacturing processes and therefore less environmental and safety concerns; 3) surface topographies can be used in conjunction with normal cleaning practices and provide an extra layer of protection. Our findings highlight the importance of exploring these alternative viral control measures and the potential benefits we could gain by evaluating other surface topographies for their efficacy against viruses.

Despite accumulating evidence that SK micropattern reduces microbial transfer, the underlying mechanism of action is not completely understood. It was proposed that three mechanisms collectively contribute for the antimicrobial efficacy of the micropattern: hydrophobicity, weakened interaction with pathogens, and reduced area for touching (see "Introduction"). However, the weight for each mechanism remains elusive. The data presented here, collected using the same method with a variety of different microorganisms, may help uncover the interplay of these mechanisms. For example, on silicone surfaces, efficacy of the micropattern ranks: *Staphylococcus aureus* > bacteriophage T4 > influenza ~ human coronavirus. We propose that such discrepancy can be explained by difference in microbial size and microbe-surface interaction. *S. aureus* (~1 μm in diameter) are considerably larger than virus particles (100–200 nm) and in the same scale as the micropattern. Therefore, the interaction between *S. aureus* and silicone surface is more drastically weakened by the micropattern. On the other hand, virion surface properties including electrostatics and hydrophobicity are known to affect their interaction with inanimate surfaces [39]. Future studies that systematically evaluate individual factors such as virion size, shape, charge, and polarity will help illustrate the detailed mechanism of how SK micropattern inhibits viral transfer on surfaces.

SK micropatterns can be applied to a wide variety of materials such as silicone, polyurethane, and polypropylene, which makes it straightforward to incorporate them into existing

products and manufacturing processes. Here we demonstrate for the first time that, when applied to different materials (silicone vs. PP), the micropattern may bring different levels of antimicrobial benefits. Such difference likely stems from the intrinsic properties of the base materials, i.e. materials that are less water repelling may benefit more from adding the micropattern, since the change in wettability would be greater. Future studies with more types of base materials would help test this hypothesis. Our findings shed light on the great potential of using SK micropatterns to limit microbial transfer on surfaces with high wettability, such as nylon, polyvinyl chloride, and polyethylene [41, 42].

## Conclusions

Infectious diseases have played a large role in shaping human history, from early plagues to recent pandemics. The toll that these diseases have taken on humanity is great, however they have also driven innovation. Many of these innovations are the hallmarks of modern society, from developing water sanitation systems to halt cholera pandemics, to cleanliness standards fighting foodborne infections, to vaccine administration eradicating smallpox and greatly reducing cases of polio and measles. It is clear that in the enduring battle between humans and pathogens, many complimentary strategies must be employed to combat both novel and historic diseases. Here, we present surface topography as one of the useful tools that can be utilized as part of a comprehensive strategy to fight disease transmission. Together with existing hygiene practices, pharmaceutical interventions, and public health policies, surface technologies can help us reduce the spread of pathogens and combat infectious diseases.

## Supporting information

**S1 Table. Measurements of *S. aureus* transfer on silicone surfaces using the bead transfer method.**
(DOCX)

**S2 Table. Measurements of T4 bacteriophage transfer on silicone surfaces using the bead transfer method.**
(DOCX)

**S3 Table. Measurements of human virus transfer on silicone surfaces using the bead transfer method.**
(DOCX)

**S4 Table. Measurements of human virus transfer on polypropylene surfaces using the bead transfer method.**
(DOCX)

## Acknowledgments

We thank Cydney Johnson, Anushila Chatterjee, and Breck Duerkop from University of Colorado Denver Anschutz Medical Campus for technical advice on bacteriophage preparation and plaque assay. We also thank Marguerite Smith and Jose Herrera for creating SK nickel masters, Hanna Kozlowski for helpful discussions during development of the beads transfer method, and Ryan Mettetal for providing the polypropylene samples.

## Author Contributions

**Conceptualization:** Qi Liu, Christopher J. Jones.

**Data curation:** Qi Liu, Lindsey Brookbank, Christopher J. Jones.

**Formal analysis:** Qi Liu.

**Funding acquisition:** Anthony B. Brennan, Christopher J. Jones.

**Investigation:** Qi Liu, Lindsey Brookbank, Angela Ho, Jenna Coffey, Christopher J. Jones.

**Methodology:** Qi Liu, Lindsey Brookbank, Angela Ho, Jenna Coffey, Christopher J. Jones.

**Project administration:** Christopher J. Jones.

**Supervision:** Anthony B. Brennan, Christopher J. Jones.

**Validation:** Qi Liu.

**Visualization:** Qi Liu, Christopher J. Jones.

**Writing – original draft:** Qi Liu, Christopher J. Jones.

**Writing – review & editing:** Qi Liu, Anthony B. Brennan, Christopher J. Jones.

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
