## [Decision Letter · Decision Letter 0]

24 Nov 2020

PONE-D-20-30941

Surface texture limits transfer of S. aureus, T4 Bacteriophage, Influenza B virus and Human coronavirus

PLOS ONE

Dear Dr. Jones,

Thank you for submitting your manuscript to PLOS ONE. After careful consideration, we feel that it has merit but does not fully meet PLOS ONE’s publication criteria as it currently stands. Therefore, we invite you to submit a revised version of the manuscript that addresses the points raised during the review process.

We look forward to receiving your revised manuscript.

Kind regards,

Amitava Mukherjee, ME, Ph.D.

Academic Editor

PLOS ONE

Journal Requirements:

'This work was funded by internal Sharklet funding. All authors were employees while working on this study. '

We note that one or more of the authors have an affiliation to the commercial funders of this research study : Sharklet Technologies, Inc.

Reviewers' comments:

Reviewer's Responses to Questions

**Comments to the Author**

1. Is the manuscript technically sound, and do the data support the conclusions?

Reviewer #1: Yes

Reviewer #2: Yes

2. Has the statistical analysis been performed appropriately and rigorously? 

Reviewer #1: Yes

Reviewer #2: Yes

3. Have the authors made all data underlying the findings in their manuscript fully available?

Reviewer #1: Yes

Reviewer #2: Yes

4. Is the manuscript presented in an intelligible fashion and written in standard English?

Reviewer #1: Yes

Reviewer #2: Yes

5. Review Comments to the Author

Reviewer #1: This manuscript described a method using bead transfer to prevent fomite transmission. The presented method is useful and adaptable to many public settings. Yet the figures are very poor, but they only provided plots which is not enough. More evidence is needed to support their conclusions, for example, CFU numbers etc. Therefore, the manuscript may be acceptable if the authors can provide more results (not just graphs) to testify the transfer in S. aureus, T4 Bacteriophage, Influenza B virus and Human coronavirus.

Reviewer #2: The manuscript deals with application of the Sharklet surface which is already a proven technology wherein inhibition of adhesion of microbes is known. However it is interesting to observe that percentage reduction of human coronavirus on these surfaces ranges from 63 -67%. The T4 bacteriophages have performed better. The data in this manuscript is of high importance and the manuscript well written. I commend the authors and strongly recommend for publication.

6. PLOS authors have the option to publish the peer review history of their article (what does this mean?). If published, this will include your full peer review and any attached files.

Reviewer #1: No

Reviewer #2: No

---

## [Author Response · Author response to Decision Letter 0]

25 Nov 2020

Reviewer #1: This manuscript described a method using bead transfer to prevent fomite transmission. The presented method is useful and adaptable to many public settings. Yet the figures are very poor, but they only provided plots which is not enough. More evidence is needed to support their conclusions, for example, CFU numbers etc. Therefore, the manuscript may be acceptable if the authors can provide more results (not just graphs) to testify the transfer in S. aureus, T4 Bacteriophage, Influenza B virus and Human coronavirus.

Response to Reviewer #1: Thank you for your comments. We have added four Supplementary Tables to the submission including the raw data for each of the microbial experiments.

Reviewer #2: The manuscript deals with application of the Sharklet surface which is already a proven technology wherein inhibition of adhesion of microbes is known. However it is interesting to observe that percentage reduction of human coronavirus on these surfaces ranges from 63 - 67%. The T4 bacteriophages have performed better. The data in this manuscript is of high importance and the manuscript well written. I commend the authors and strongly recommend for publication.

Response to Reviewer #2: Thank you for the positive comments regarding the manuscript. We agree that this is an important finding in the struggle against surface contamination across a wide range of infectious agents.

---

## [Decision Letter · Decision Letter 1]

11 Dec 2020

Surface texture limits transfer of S. aureus, T4 Bacteriophage, Influenza B virus and Human coronavirus

PONE-D-20-30941R1

Dear Dr. Jones,

We’re pleased to inform you that your manuscript has been judged scientifically suitable for publication and will be formally accepted for publication once it meets all outstanding technical requirements.

Kind regards,

Amitava Mukherjee, ME, Ph.D.

Academic Editor

PLOS ONE

Additional Editor Comments (optional):

Reviewers' comments:

Reviewer's Responses to Questions

**Comments to the Author**

1. If the authors have adequately addressed your comments raised in a previous round of review and you feel that this manuscript is now acceptable for publication, you may indicate that here to bypass the “Comments to the Author” section, enter your conflict of interest statement in the “Confidential to Editor” section, and submit your "Accept" recommendation.

Reviewer #2: All comments have been addressed

2. Is the manuscript technically sound, and do the data support the conclusions?

Reviewer #2: Yes

3. Has the statistical analysis been performed appropriately and rigorously? 

Reviewer #2: Yes

4. Have the authors made all data underlying the findings in their manuscript fully available?

Reviewer #2: Yes

5. Is the manuscript presented in an intelligible fashion and written in standard English?

Reviewer #2: Yes

6. Review Comments to the Author

Reviewer #2: The addition of the tables and raw data supplement the findings. The manuscript is accepted for publication.

7. PLOS authors have the option to publish the peer review history of their article (what does this mean?). If published, this will include your full peer review and any attached files.

Reviewer #2: No

---

## [Editor Report · Acceptance letter]

15 Dec 2020

PONE-D-20-30941R1 

Surface texture limits transfer of S. *aureus*, T4 Bacteriophage, Influenza B virus and Human coronavirus. 

Dear Dr. Jones:

I'm pleased to inform you that your manuscript has been deemed suitable for publication in PLOS ONE. Congratulations! Your manuscript is now with our production department. 

Kind regards, 

on behalf of

Professor Dr. Amitava Mukherjee 

Academic Editor

PLOS ONE